# Knowing How to Edit: Reliable Evaluation Signals for Diagnosing and Optimizing Prompts at Query Level

## Abstract

Prompt optimization has become a central mechanism for eliciting strong performance from LLMs, and recent work has made substantial progress by proposing diverse prompt evaluation metrics and optimization strategies. Despite these advances, prompt evaluation and prompt optimization are often developed in isolation, limiting the extent to which evaluation can effectively inform prompt refinement. In this work, we study prompt optimization as a process guided by performance-relevant evaluation signals. To address the disconnect between evaluation and optimization, we propose an evaluation-instructed prompt optimization approach that explicitly connects prompt evaluation with query-dependent optimization. Our method integrates multiple complementary prompt quality metrics into a performance-reflective evaluation framework and trains an execution-free evaluator that predicts prompt quality directly from text, avoiding repeated model executions. These evaluation signals then guide prompt refinement in a targeted and interpretable manner. Empirically, the proposed evaluator achieves 83.7% accuracy in predicting prompt performance. When incorporated into the optimization process, our approach consistently outperforms existing optimization baselines across eight benchmark datasets and three different backbone LLMs. Overall, our results demonstrate that reliable and efficient evaluation signals can serve as an effective foundation for robust and interpretable prompt optimization.

## 1 Introduction

As the capabilities of large language models (LLMs) continue to improve, prompting has become a primary interface for eliciting and shaping model behavior Liu et al. (2025). This has sparked rapid progress along two closely related directions: methods for evaluating prompt quality, and strategies for optimizing prompts to improve downstream performance. A wide range of prompt evaluation metrics and optimization techniques have been proposed, reporting notable performance improvements across diverse tasks. Despite this progress, prompt evaluation and prompt optimization are typically developed as separate components. Many evaluation methods focus on measuring prompt quality without indicating how prompts should be modified, while optimization procedures frequently rely on heuristic, unstable, or opaque signals that are weakly grounded in systematic evaluation. This separation raises a fundamental question: **How can evaluation provide reliable signals for deciding where and how to refine prompts?**

In this work, we take a step toward connecting prompt evaluation with prompt optimization. Rather than treating evaluation as a post-hoc assessment, we explore how evaluation signals can be used to inform where and how prompts should be refined. At a high level, this perspective involves two closely related aspects: (i) how prompt quality should be comprehensively evaluated, and (ii) how such evaluation signals can be used to guide prompt optimization.

Existing prompt optimization strategies broadly fall into two categories: static template optimization and dynamic, query-level optimization. Traditional approaches optimize a single prompt template for a given task and apply it uniformly to all queries, as in methods such as *TextGrad* Yuksekgonul et al. (2024), *ProTeGi* Pryzant et al. (2023), and their variants. These methods implicitly assume that queries within a task share similar semantic and reasoning structures, allowing a global template to generalize. However, as LLM applications expand to more diverse and open-ended settings, this assumption becomes increasingly fragile. In dynamic and complex real-world user scenarios, static templates often fail to adapt, leading to suboptimal or unstable performance.

To address this limitation, recent work has moved toward query-dependent prompt optimization, including methods such as *IAP* Yuan et al. (2024), *Self-Refine* Madaan et al. (2023), *QPO* Kong et al. (2024), and *Prompt-OIRL* Sun et al. (2023a). These approaches tailor prompts to individual queries rather than relying on a single template. Despite their flexibility, most dynamic methods rely on heuristic, noisy, or opaque optimization signals, such as LLM-generated textual feedback or black-box reward models. As a result, optimization is often unstable and difficult to interpret, as these methods lack a principled way to decide where and how prompts should be edited. This highlights the need for reliable evaluation signals to guide prompt optimization.

A natural solution for providing such guidance is prompt evaluation. A growing body of work has proposed diverse metrics to assess prompt quality, including semantic LLM-based measures and quantitative, response-based metrics such as *stability* Chen et al. (2025) and *mutual-information* scores Kraskov et al. (2004). While these metrics offer useful perspectives, they are typically developed in isolation, capturing only partial aspects of prompt behavior and exhibiting weak or inconsistent alignment with downstream task performance. The absence of a systematic, performance-reflective evaluation framework has therefore limited their use as actionable signals for prompt optimization, and prompt evaluation is often treated as a post-hoc assessment rather than as a tool for guiding prompt refinement. Moreover, many evaluation metrics are response-based and require repeated model executions, making them difficult to deploy in query-level or multi-agent settings.

To address these challenges, we propose an evaluation-instructed prompt optimization system that explicitly connects prompt evaluation with optimization decisions. Rather than treating evaluation as a post-hoc analysis, our goal is to use evaluation signals to determine where and how prompts should be edited during optimization.

First, to overcome the fragmented nature of existing metrics, we construct a performance-reflective evaluation framework that integrates multiple complementary dimensions of prompt quality and aligns them with downstream task accuracy. Second, to mitigate the high cost of response-based evaluation, we train an execution-free evaluator that predicts prompt quality scores directly from text, enabling prompt assessment without repeated model executions. Finally, we use these evaluation signals to guide prompt optimization, allowing prompt revisions to be targeted, interpretable, and query-dependent.

Empirically, the proposed evaluator achieves an accuracy of 83.7% in predicting prompt success rate. When integrated with the metric-aware optimization process, our approach consistently outperforms both static-template optimization methods and existing query-dependent optimization baselines across 8 benchmark datasets on 3 different backbone LLMs.

Our main contribution is proposing **the first prompt evaluation–optimization pipeline**, which unifies comprehensive evaluation and dynamic optimization within a single closed loop. Building on this pipeline, our contributions are twofold:

1. **Performance-reflective prompt evaluation.** We establish a comprehensive prompt-quality evaluation system, and train an execution-free evaluator capable of predicting multi-dimensional quality and downstream performance, which provides reliable signal for prompt optimization.
2. **Evaluation–instructed optimization mechanism.** We bridge the gap between prompt evaluation and optimization in query-dependent settings, improving robustness and interpretability by allowing evaluation signals to directly guide optimization directions.

## 2 Related Works

### 2.1 Static Prompt Template Optimization

A substantial body of prior work studies prompt optimization by searching for a single prompt template that generalizes across all queries within a task. Representative approaches include Automatic Prompt Engineer (APE) Zhou et al. (2023), OPRO Yang et al. (2024), and PromptAgent Wang et al. (2023). These methods typically generate candidate prompts, execute them on a training set, evaluate performance against ground-truth responses, and iteratively refine prompts using reinforcement learning, evolutionary strategies, or heuristic search. Such approaches have been shown to automate prompt discovery and improve performance over manually designed templates.

However, static template optimization assumes that queries within a task share sufficiently similar linguistic and reasoning structures for a single prompt to generalize. In practice, real-world queries often exhibit substantial heterogeneity. For example, reasoning benchmarks such as GSM8K Cobbe et al. (2021) contain problems that differ widely in both surface form and underlying logical structure. Even an optimized static prompt primarily encodes task-level instructions and cannot adapt its reasoning strategy or contextual emphasis to individual queries Sun et al. (2023b). As a result, the effectiveness of static prompts is fundamentally constrained in settings where query-level variation is pronounced.

From a modeling perspective, static prompt optimization can be viewed as learning a global prompt that minimizes average loss over a task distribution. While effective in relatively homogeneous settings, this formulation limits the ability of prompts to exploit query-specific structure or contextual information, motivating the development of more adaptive optimization strategies.

### 2.2 Query-Dependent Prompt Optimization

To overcome the limitations of static templates, recent work has explored query-dependent prompt optimization, where prompts are tailored to individual queries. Instance-Adaptive Prompting (IAP) Yuan et al. (2024), for example, selects prompts based on attention-gradient saliency scores derived from internal model representations. Self-Refine Madaan et al. (2023) and ProRefine Pandita et al.

(2025) iteratively improve prompts using LLM-generated textual feedback, while QPO Kong et al. (2024) and Prompt-OIRL Sun et al. (2023a) train auxiliary models to generate or rank prompts for each instance.

Although these methods increase flexibility and enable query-specific adaptation, they differ substantially in how optimization signals are obtained. Many approaches, such as Self-Refine Madaan et al. (2023) and ProRefine Pandita et al. (2025), rely on natural-language feedback generated by LLMs to approximate optimization directions, which can introduce instability and sensitivity to sampling noise Gou et al. (2023). Others depend on black-box reward models or access to internal model states (e.g., IAP Yuan et al. (2024), QPO Kong et al. (2024), and Prompt-OIRL Sun et al. (2023a) ), which limits interpretability, portability, or scalability. As a result, while query-dependent methods improve expressiveness compared to static templates, they often lack clear and interpretable mechanisms for determining how prompts should be modified.

## 2.3 Prompt Evaluation Metrics

Parallel to advances in prompt optimization, a growing literature has proposed metrics for evaluating prompt quality. Existing metrics span semantic LLM-based measures, such as clarity, coherence, and specificity Shah (2024), as well as quantitative, response-based metrics including stability Chen et al. (2025) and mutual-information scores Kraskov et al. (2004). These metrics provide complementary perspectives on prompt behavior, capturing aspects such as semantic alignment, output consistency, and sensitivity to sampling.

However, most prompt evaluation metrics are developed independently and focus on one specific attribute of prompt behavior, leading to fragmented assessments that may not consistently reflect downstream task performance. Moreover, many commonly used metrics, such as stability Chen et al. (2025), MI Kraskov et al. (2004), and prompt entropy Lu et al. (2022), are response-based and require repeated executions of an LLM to obtain stable estimates, which significantly increases computational cost. This limits their practical use in settings where prompts must be evaluated or updated at the query level, such as interactive systems or multi-agent frameworks.

Due to these limitations, prompt evaluation is often used as a post-hoc analysis tool rather than as a direct input to prompt optimization. Bridging prompt evaluation with optimization in a principled and efficient manner remains an open challenge.

## 3 Method

Our goal is to build a reliable and interpretable prompt evaluator that can estimate prompt quality across multiple dimensions without executing the prompt itself, and use the scores on different metrics to instruct the prompt optimization process. This section introduces how we construct the training dataset, select quantitative evaluation metrics, design and train the evaluator model, and use the evaluation results to instruct query dependent prompt optimization.

### 3.1 Training Dataset Generation

To train an evaluator that accurately captures how prompts behave across different quality dimensions, we require a dataset that spans a wide spectrum of prompt quality. Rather than generating only high-performing prompts, our goal is to construct a diverse collection of prompts with varying

structures, styles, and performance levels. This diversity enables the evaluator to learn generalizable patterns relating prompt characteristics to downstream execution performance. In other words, the objective of this stage is *diversity*, not optimality; main performance improvement is achieved during the subsequent optimization stage (Section 3.4).

We sample queries from multiple benchmarks, including BBH Suzgun et al. (2022) (*causal_judgement*, *disambiguation_qa*, *sports_understanding*, *web_of_lies*), GPQA Rein et al. (2024) (*diamond*, *main*), and LegalBench Guha et al. (2023) (*definition_classification*). For each query, we generate prompt candidates from three complementary sources:

**(1) Static prompt templates.** We select five widely used prompt templates covering several mainstream prompting paradigms, including direct prompting, chain-of-thought, tree-of-thought, and multi-expert debate/voting structures. The templates are shown in Appendix Table 5, which serve as a diverse and stylistically rich foundation for prompt construction.

**(2) LLM-generated prompts.** To introduce systematic stylistic diversity, we instruct the LLM with a higher temperature to generate new prompts conditioned on each query. We design six prompting styles: *step-by-step reasoning, expert discussion, Socratic questioning, creative exploration, verification-oriented prompting*, and *contrastive prompting*. We provide concrete examples of these six prompting styles in Appendix Table 6.These styles reflect different reasoning dynamics and linguistic patterns, enabling the evaluator to observe a broad set of semantic variations.

**(3) Evolutionary recombination.** Inspired by genetic algorithms, we further expand the prompt pool through semantic recombination. From the union of static and LLM-generated prompts, we randomly sample two parent prompts. We apply an LLM agent to semantically decompose each parent prompt into two segments, cross-combine the segments into new hybrids, and then rephrase the resulting prompts to ensure smoothness and semantic coherence. Illustrative examples of the recombination process are provided in Appendix Table 7 This procedure effectively blends structural and stylistic features from multiple prompting strategies while maintaining natural-language fluency.

Across these three sources, we obtain a total of 11,530 prompt candidates with substantial diversity in reasoning structure, linguistic form, semantic content, and overall quality. This diverse dataset provides a strong foundation for training a generalizable evaluator.

### 3.2 Metric Selection

To comprehensively evaluate prompt quality, we collect a diverse set of quantitative metrics commonly used in existing prompt-evaluation research. These metrics can be grouped into four categories: (1) **LLM-based semantic metrics**: *clarity, coherence*, and *specificity* Shah (2024); (2) **Prompt-intrinsic metric**: *nll_score* Lastras (2019); (3) **Response-based metrics**: *stability_score* Chen et al. (2025), *mi_score* Kraskov et al. (2004), and *prompt_entropy* Lu et al. (2022), each computed from multiple executions of the same prompt; (4) **Task-level metric**: *query_entropy* Lu et al. (2022), which captures the inherent uncertainty or difficulty of the query itself.

In addition to these metrics, we record each prompt's execution accuracy. To obtain stable estimates under stochastic decoding, every prompt is executed ten times. We then use whether its average accuracy exceeds 50% as a binary indicator of prompt quality. This threshold follows a majority-voting logic: once a prompt's accuracy exceeds 50%, aggregating multiple executions by majority vote will consistently recover the correct answer despite sampling noise.

Using all 11,530 prompts, we first compute their full multi-dimensional metric scores strictly following the formal definitions of each metric. For metrics that require model execution (e.g., response-based metrics), we execute each prompt ten times under controlled sampling settings and average the resulting scores to obtain stable estimates. With these rigorously obtained metric scores, we then perform metric selection based on their ability to reflect downstream performance. Specifically, we generate semantic embeddings for each prompt using the `text-embedding-3-large` model, train an XGBoost classifier to predict whether accuracy exceeds 50%, and use the gain importance scores to evaluate each metric's contribution. Metrics with only weak associations to performance are removed.

Through this analysis, we identify four metrics that exhibit both high importance and strong complementarity. Each metric captures a distinct dimension of prompt behavior:

(1) **nll_score**: Measures the negative log-likelihood of directly generating the correct answer. Lower scores indicate stronger semantic guidance and a more restrictive output space.

(2) **stability_score**: Quantifies response consistency across repeated executions. Higher stability reflects reduced sensitivity to sampling noise.

(3) **mi_score**: Computes the mutual information between the query and the generated output, representing semantic alignment and task relevance.

(4) **query_entropy**: Captures the intrinsic difficulty of the query by accessing the entropy. Ambiguous or high-reasoning queries exhibit higher entropy without prompting.

These four metrics were used both as supervision targets and as interpretability anchors for the evaluator, jointly providing a comprehensive and complementary view of prompt quality. The detailed formulas and underlying rationale for each metric will be further discussed in Section 3.4.

### 3.3 Evaluator Architecture

Our evaluator $\mathcal{E}_\theta$ is built upon a LLaMA-8B-Instruct backbone with lightweight LoRA adaptation, and is designed as a multi-task model that jointly performs binary classification and metric regression. Formally, the evaluator implements a mapping:

$$\mathcal{E}_\theta : x \mapsto (\hat{y}, \hat{\mathbf{m}}), \quad \text{with} \quad \hat{y} = f_{\text{cls}}(z; \theta_c), \quad \hat{\mathbf{m}} = f_{\text{reg}}(h; \theta_r) \in \mathbb{R}^M.$$

Here, $h$ denotes the shared encoder representation, and $z$ is the metric-aware fusion of semantic and regression features. Both quantities are formally defined in the following equations.

**Shared Encoder.** The evaluator constructs the textual input from the user query $q$ and the prompt candidate $p$. To exploit the model's reasoning capabilities, we prepend a fixed natural-language prefix $r$ describing the evaluation objective and the definition of each metric; this improves stability and ensures the encoder internalizes the meaning of every score dimension. The

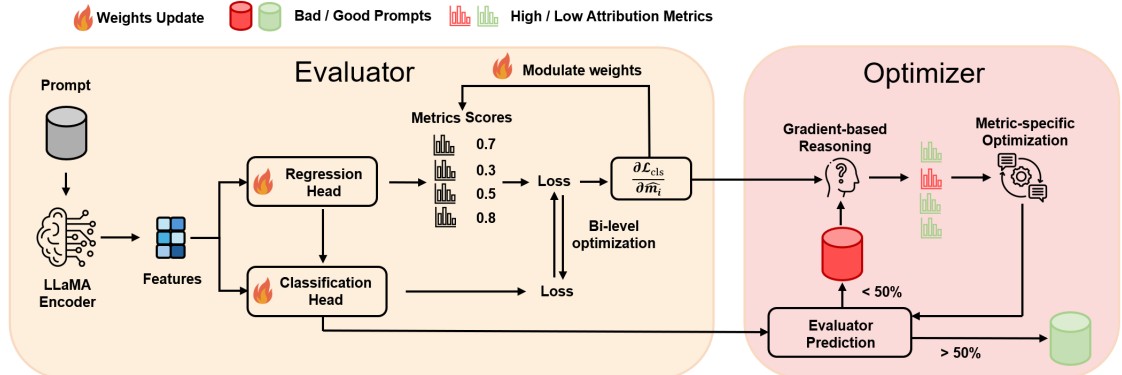

Figure 1: Overview of our evaluation-instructed optimization pipeline.

concatenated input is then encoded through the LLaMA backbone:

$$x = [r; \ q; \ p], \qquad h = f_{\text{enc}}(x; \theta_{\text{enc}}).$$

**Metric regression and feature fusion.** A regression head predicts continuous values for the selected metrics. To enable the classifier to be both semantics-aware and metric-aware, the predicted metric vector is fused with the feature map via a lightweight MLP:

$$z = \phi(h, \ \hat{\mathbf{m}}) = MLP([h; \hat{\mathbf{m}}]).$$

This fusion encourages the classifier to base its decision not only on prompt semantics but also on the evaluator's internal expectations of metric behavior.

**Bi-level training objective.** The evaluator is trained through a bi-level objective in which regression assists classification by providing additional structural signals. For a fixed set of metric weights $\mathbf{w}$, the model parameters $\theta$ are optimized through:

$$\theta^*(\mathbf{w}) = \arg\min_{\theta} \left( \mathcal{L}_{\text{cls}}(\theta) + \lambda \sum_{i=1}^{M} w_i \, \mathcal{L}_i^{\text{reg}}(\theta) \right), \qquad \mathbf{w}^* = \arg\min_{\mathbf{w}} \mathcal{L}_{\text{cls}}(\theta^*(\mathbf{w})) \, .$$

This design ensures that metric regression helps classification only when useful, preventing the regression task from dominating or misguiding learning.

**Gradient-informed metric weighting.** To identify which metric dimensions most strongly affect prompt success, we update each weight $w_i$ using gradient signals from the classification loss:

$$w_i \leftarrow \mathcal{A}(w_i, \ \|\nabla_{\hat{m}_i} \mathcal{L}_{\text{cls}}\|) \quad \text{with} \quad \mathcal{A}(w_i, g_i) = Normalize(w_i - \eta \, \frac{\partial \mathcal{L}_{\text{cls}}}{\partial \hat{m}_i}),$$

where the adaptation operator increases the influence of metrics that exert stronger gradients on the classification objective. This creates a closed-loop mechanism in which regression dimensions are automatically aligned with performance-relevant signals.

**Lightweight adaptation.** To maintain efficiency and portability, we fine-tune only a small subset of attention layers through LoRA adapters, amounting to roughly 5.5% of total parameters. This keeps computational cost low while preserving generalization across prompt types and downstream models.

### 3.4 Metric-Aware Optimization

Once the evaluator identifies a prompt as low-quality (i.e., $\hat{y} < 0.5$), the system initiates a metric-instructed optimization process. The core idea is to trace *why* a prompt fails by examining the gradient of the classification loss with respect to each predicted metric dimension, which quantifies the negative contribution to the evaluator's decision of a particular metric. This enables the system to attribute performance degradation to specific factors (e.g., low answer confidence, high ambiguity, unstable reasoning trajectory, ect.) rather than treating all errors as a single homogeneous failure mode. For each problematic metric dimension, the optimizer invokes a corresponding diagnostic module that inspects the prompt for metric-specific issues and suggests targeted corrective edits. These refined suggestions are aggregated to construct an improved prompt, which is then reevaluated.

Below we present the four core metrics used in this process, including their definitions, diagnostic intuition, and how the optimization module improves each dimension.

**Negative Log-Likelihood (NLL).** For a query–prompt pair $(q, p)$ and the correct answer token sequence $\mathbf{y} = (y_1, \ldots, y_T)$, the NLL score is computed by forcing the model to generate the correct answer and measuring the probability:

$$\text{NLL}(p, q) = -\frac{1}{T} \sum_{t=1}^{T} \log P_\theta(y_t \mid q, p, y_{<t}).$$

In practice, this measures the model's confidence in the correct answer. A high NLL (i.e., low confidence) suggests that the prompt does not sufficiently constrain the model's generative distribution.

Typical causes of poor NLL include: (1) *Instruction conflict*: the prompt simultaneously demands incompatible goals, diluting the model's focus; (2) *Noisy or long preambles*: excessive role-play, fillers, or redundant restatements weaken the clarity of the task; (3) *Few-shot inconsistency*: demonstrations differ in reasoning style or label format, confusing the model's internal alignment; (4) *Task mismatch*: the reasoning template implied by the prompt is misaligned with the actual task type. A metric-specific diagnoser then detects these issues and provide optimization suggestions.

**Semantic Stability.** For a prompt $p$, we sample $N$ model outputs $a_1, \ldots, a_N$ and embed each output via $v_i = \phi(a_i)$. Stability score is calculated as:

$$S(p, q) = 1 - \frac{2}{N(N-1)} \sum_{i<j} d_{ij}, \quad with \quad d_{ij} = 1 - \frac{v_i \cdot v_j}{\|v_i\| \, \|v_j\|}.$$

High stability implies more stable and robust responses.

Instability typically arises from: (1) *Unspecified output format*: missing a deterministic "final answer" field or clear parseable structure; (2) *Conflicting objectives*: prompts encouraging creativity or multiple perspectives during an accuracy-oriented task; (3) *Unconstrained reasoning paths*: allowing arbitrary-length or hedge-heavy reasoning chains introduces randomness; (4) *Missing guiding example*: the lack of a stable template increases trajectory drift. The optimizer therefore introduces fixed-slot formats, removes diversity-inducing instructions, constrains reasoning depth, or uses micro few-shot examples.

**Mutual Information (MI).**    We measure how strongly the prompt influences the model's output beyond the query alone:

$$\mathrm{MI}(p,q) = H(A \mid q) - H(A \mid q,p), \qquad H(A \mid q,p) = -\sum_a \hat{P}(a \mid q,p) \log \hat{P}(a \mid q,p).$$

High MI indicates that the prompt provides meaningful, actionable guidance; low MI suggests that the output is nearly independent of the prompt.

Low MI is usually caused by: (1) *Hollow templates*: vague instructions like "answer carefully" that lack concrete operational cues; (2) *Stylistic noise*: role-play, persona, or politeness instructions that add tokens but not guidance; (3) *Missing schemas*: prompts that do not define variables, conditions, or verification steps. Optimization introduces explicit variable definitions, checklist-style schemas, and strong reasoning cues that increase prompt–response coupling.

**Query Entropy.**    To measure the intrinsic difficulty of the query itself, we estimate the entropy of the model's answer distribution when no prompt is provided:

$$H(A \mid q) = -\sum_a \hat{P}(a \mid q) \log \hat{P}(a \mid q).$$

Higher entropy indicates that the query naturally induces divergent or unstable answers, often because the problem lacks explicit assumptions or requires domain-specific guidance.

Common sources of high query entropy include: (1) *Ambiguity or missing assumptions*: unclear definitions, scope, or hidden constraints; (2) *Lack of reasoning structure*: the prompt does not provide a stable scaffold for step-by-step solving; (3) *Missing domain context*: specialized tasks require minimal technical framing that the prompt may not supply; (4) *Unconstrained output space*: no guidance on allowable answer formats or numeric ranges. For the query-entropy dimension, the optimizer first invokes ambiguity- and uncertainty-focused diagnosers to detect potential issues. Unlike the other metrics, whose fixes modify the system prompt, improvements here are applied by augmenting the query side with minimal clarifications and constraints.

**Gradient-Based Attribution and Revision.**    For each metric $\hat{m}_i$, the evaluator computes

$$g_i = \left\| \nabla_{\hat{m}_i} \mathcal{L}_{\mathrm{cls}} \right\|,$$

which quantifies how strongly that metric dimension influences the "bad" classification. These sensitivities guide both the reweighting of regression losses and the selection of metric-specific

diagnostic prompts. The optimizer then queries the corresponding metric-specific diagnosers (e.g., instruction-conflict detector for NLL, format guard for stability, etc,) and rewrites the prompt dimension-by-dimension.

By grounding optimization in interpretable and metric-aligned signals from the evaluator, the overall procedure becomes more stable, principled, and effective than heuristic prompt rewriting approaches.

## 4 Experiments

### 4.1 Setup

Our experiments cover both open-domain and domain-specific datasets, including the BBH tasks Suzgun et al. (2022) (*causal judgement*, *disambiguation QA*, *sports understanding*, *web of lies*), *GPQA Diamond* Rein et al. (2024), *LegalBench definition classification* Guha et al. (2023), *MATH500* Lightman et al. (2023), and the medical QA dataset *MedQA* Jin et al. (2021). These tasks span diverse reasoning types—causal inference, semantic disambiguation, common-sense reasoning, factual judgment, and professional knowledge QA—providing a broad evaluation ground for prompt performance across varied semantic and reasoning settings.

For each dataset, we randomly sample 100 examples for training and 100 for testing; for datasets with fewer than 200 samples, we adopt a 50%–50% train–test split. Using the three diverse prompt-generation strategies described in Section 3.1, we generate multiple prompt samples for each question and construct a prompt pool containing 11,530 examples. Among these, *MedQA* Jin et al. (2021) and *MATH500* Lightman et al. (2023) are excluded from training and used exclusively to evaluate the method's generalization ability to **unseen tasks**, especially in specialized domains such as medicine.

For model setup, we use LLaMA-3-8B-Instruct as the sole base model for generating training data and training the evaluator. During testing, we used LLaMA-3-8B-Instruct, LLaMA-3.1-8B-Instruct, and GPT-4o as prompt execution backbones to examine whether an evaluator trained on a single model can generalize across different LLMs.

Notably, the "training" process here refers solely to training the evaluator—learning to predict prompt quality across different metrics and execution performance. The optimization process is untrained, relying entirely on the evaluator's predicted scores to guide dynamic prompt adjustment.

### 4.2 Metric Selection Results

We evaluated eight widely used candidate metrics: *clarity*, *coherence*, *specificity*, *nll_score*, *stability_score*, *mi_score*, *prompt_entropy*, and *query_entropy*. These metrics were used as input features to an XGBoost classifier predicting whether each prompt's execution accuracy exceeded 50%. Table 1 shows the feature importance distribution of all metrics. Following a threshold of overall gain contribution greater than 10%, we selected four core metrics that are most strongly correlated with downstream performance: **nll_score**, **stability_score**, **mi_score**, and **query_entropy**.

These features are complementary and relatively independent in terms of information contribution. Although XGBoost does not explicitly model orthogonality among features, its layer-wise decision structure implicitly reduces the importance of collinear variables once key features have

Table 1: Feature importance of prompt evaluation metrics from XGBoost.

| Feature | Importance | Weight (%) |
|---|---|---|
| query_entropy | 0.101 | 30.9 |
| stability_score | 0.066 | 20.2 |
| nll_score | 0.043 | 13.0 |
| mi_score | 0.035 | 10.7 |
| prompt_entropy | 0.025 | 7.5 |
| specificity | 0.022 | 6.8 |
| clarity | 0.020 | 6.1 |
| coherence | 0.016 | 4.9 |

Table 2: Learned weights of prompt evaluation metrics from our evaluator.

| Feature | Weight (%) |
|---|---|
| query_entropy | 32.7 |
| nll_score | 26.4 |
| stability_score | 22.3 |
| mi_score | 18.6 |

been explained, thereby minimizing redundancy. For example, the *mutual information* score partially captures the semantic diversity represented by *prompt entropy*, which explains its higher relative importance and the natural attenuation of *prompt entropy*. Overall, this mechanism helps preserve feature complementarity and enables multi-dimensional modeling of prompt quality and performance.

### 4.3 Metric and Performance Prediction

Based on the prompt pool constructed in Section 3.1 and the selected performance-reflective metrics, we fine-tune the evaluator model. 80% of samples are used for training and 20% for validation. On the validation set, our evaluator achieves a prompt-quality classification accuracy of **83.7%**, significantly outperforming random and baseline levels. For comparison, we tested the evaluation model in Prompt-OIRL Sun et al. (2023a), which is based on embedding + XGBoost. Under identical data and evaluation settings, even when provided with ground-truth metric scores, it achieves only **69%** accuracy—indicating that our evaluator more effectively captures the relationship between prompt semantics and execution performance.

**Ablation Study.** To assess the contribution of each component, we conduct systematic ablation experiments by removing or replacing key modules: (1) using all metrics instead of the selected performance-related ones; (2) removing the task-descriptive prefix from prompts; (3) removing the guidance from predicted metric regression results to the classification task; (4) removing the fusion of metric scores with the encoder's feature map; and (5) disabling the gradient-based dynamic weighting of regression losses.

The results demonstrate that: (1) keeping only representative metrics reduces noise and redundancy, improving learning efficiency; (2) providing task-descriptive prefixes enhances the evaluator's semantic understanding; (3) fusing metric predictions with encoder features improves the model's ability to assess prompt quality; and (4) dynamically weighting metric losses according to classification gradients is beneficial, as different metrics contribute unequally to final performance.

We further extract the learned weights of each metric after training. As shown in Table 2, the distribution remains consistent with the XGBoost results—dominated by **query_entropy** (32.7%), followed by **nll_score**, **stability_score**, and **mi_score**. This finding indicates that the effect of prompts on LLM performance is largely bounded by the intrinsic difficulty of the query and the

Table 3: Ablation results of the evaluator model.

| Configuration | Validation Accuracy |
|---|---|
| Complete evaluator | **83.7%** |
| Use all metrics | 79.6% |
| w/o prefix | 80.2% |
| w/o predicted metric score | 75.5% |
| w/o feature map from encoder | 67.3% |
| w/o dynamic loss weighting | 80.6% |

capability ceiling of the model. This is expected: the purpose of prompt optimization is not to enable a weaker model to achieve a qualitative leap (e.g., making LLaMA-8B outperform GPT-4o), but rather to maximize its potential within existing capability limits. In other words, while prompt optimization cannot alter the model's capacity boundary, it can yield substantial gains within that boundary. Compared with XGBoost, our evaluator exhibits a more balanced weighting across metrics and significantly higher overall accuracy, suggesting that it learns richer and more complementary information from multiple dimensions.

## 4.4 Systematic Performance

We compare our framework with two major categories of prompt optimization baselines: (1) **Query-dependent optimization** methods: *Self-Refine* and *Pro-Refine*; and (2) **Static template optimization** methods: *APE* Zhou et al. (2023) and *TextGrad*. For fairness, all methods are evaluated on the same base model, LLaMA-3-8B-Instruct, with the maximum number of optimization iterations set to 3.

As shown in Table 4, our method achieves the best performance on nearly all tasks. Notably, for all three different backbone LLMs, it attains approximately 10% improvement over the single-agent baseline on the *LegalBench definition classification* task and a 5%-6% gain on *MedQA*, an unseen medical-domain task. These results demonstrate that our framework consistently enhances performance across seen tasks and generalizes effectively to new domains. This generalization is reasonable, as our training focuses solely on learning the evaluator's scoring ability rather than fitting to specific tasks or domains—thus providing transferable optimization signals even for unseen scenarios.

Table 4 also demonstrated the cross-model generalization of our system. Although all training samples are generated from LLaMA-3-8B-Instruct, we replace it with LLaMA-3.1-8B-Instruct and GPT-4o during testing as the prompt execution models. Figure 2 visualizes the performance improvements of our optimized prompts over the LLM-only baseline on three backbone models (LLaMA-3, LLaMA-3.1, and GPT-4o). Each bar shows the relative gain (or drop) for a given dataset–model pair, with green indicating improvements and red indicating performance decreases. We observe that, despite being trained exclusively on LLaMA-3-8B samples, our evaluator consistently produces positive gains across most datasets and models. The overall pattern remains stable across backbones, suggesting that the learned metric-aware optimization signals are highly transferable and largely model-agnostic. Remaining variations mainly reflect differences in backbone capability rather than failures of the optimization mechanism.

| Task | Model | LLM only | Query-dependent Optimization | | | Template Optimization | |
| | | | Ours | Self-Refine | Pro-Refine | TextGrad | APE |
|---|---|---|---|---|---|---|---|
| bbh_causal judgement | llama3 | 0.60 | **0.63** | 0.58 | 0.61 | 0.62 | 0.59 |
| | llama3.1 | 0.58 | **0.64** | 0.59 | 0.61 | 0.57 | 0.53 |
| | gpt-4o | 0.73 | **0.78** | 0.74 | 0.75 | 0.74 | 0.72 |
| bbh_disambi- guation_qa | llama3 | 0.63 | **0.65** | 0.65 | 0.63 | 0.64 | 0.63 |
| | llama3.1 | 0.64 | 0.65 | **0.66** | 0.64 | 0.65 | 0.63 |
| | gpt-4o | 0.58 | 0.69 | 0.56 | 0.63 | **0.71** | 0.67 |
| bbh_sports understanding | llama3 | 0.68 | **0.75** | 0.68 | 0.71 | 0.70 | 0.69 |
| | llama3.1 | 0.67 | **0.75** | 0.71 | 0.70 | 0.73 | 0.68 |
| | gpt-4o | 0.78 | **0.83** | 0.79 | 0.82 | 0.80 | 0.79 |
| bbh_web of_lies | llama3 | 0.66 | 0.69 | 0.67 | 0.71 | **0.74** | 0.72 |
| | llama3.1 | 0.69 | 0.68 | 0.66 | 0.70 | **0.73** | 0.71 |
| | gpt-4o | 0.96 | **0.98** | 0.94 | 0.96 | 0.96 | 0.95 |
| GPQA Diamond | llama3 | 0.28 | **0.29** | 0.26 | 0.28 | 0.26 | 0.27 |
| | llama3.1 | 0.26 | 0.27 | 0.26 | **0.29** | 0.27 | 0.24 |
| | gpt-4o | 0.22 | **0.24** | 0.20 | 0.21 | 0.23 | 0.22 |
| Legal Bench | llama3 | 0.55 | **0.70** | 0.63 | 0.63 | 0.58 | 0.55 |
| | llama3.1 | 0.56 | **0.69** | 0.63 | 0.63 | 0.58 | 0.61 |
| | gpt-4o | 0.83 | **0.90** | 0.81 | 0.86 | 0.84 | 0.84 |
| MATH500 | llama3 | 0.31 | – | – | – | – | – |
| | llama3.1 | 0.36 | – | – | – | – | – |
| | gpt-4o | 0.81 | **0.86** | 0.78 | 0.80 | 0.81 | 0.82 |
| MedQA | llama3 | 0.47 | **0.52** | 0.50 | 0.51 | 0.50 | 0.49 |
| | llama3.1 | 0.46 | **0.51** | 0.49 | 0.51 | 0.48 | 0.44 |
| | gpt-4o | 0.51 | **0.57** | 0.54 | 0.46 | 0.49 | 0.51 |

Table 4: Performance comparison with baseline prompt-optimization methods across three backbone LLMs (LLaMA-3, LLaMA-3.1, GPT-4o). Results for MATH500 are omitted for LLaMA-3/3.1 because these mid-sized models lack sufficient arithmetic capability, causing errors dominated by computation rather than prompt design.

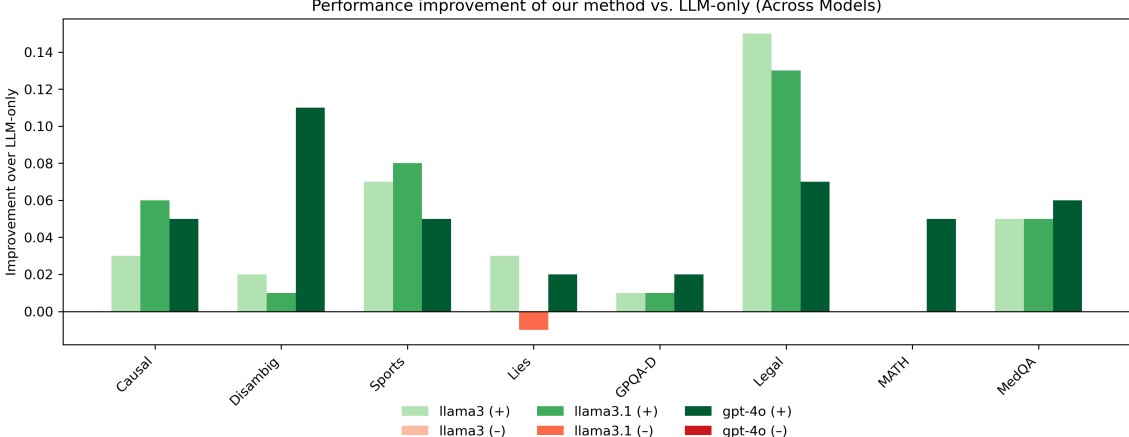

Figure 2: Improvement of our optimized prompts over the LLM-only baseline across three backbone models (LLaMA-3, LLaMA-3.1, and GPT-4o). Positive gains are shown in green and negative changes in red, with color intensity reflecting model size.

This stable cross-model performance highlights the core advantage of our framework: by decomposing prompt quality into interpretable multi-dimensional metrics and forming a closed evaluation–optimization loop, our method achieves consistent, efficient, and robust prompt optimization across diverse tasks and backbone models.

## 5 Conclusion

This work presents a systematic and interpretable prompt evaluation–optimization framework that unifies multi-dimensional evaluation and dynamic optimization within a single closed loop. Unlike prior approaches that either rely on textual feedback or black-box reward models, our framework establishes a metric-based evaluator that predicts prompt quality directly from text without execution, bridging the gap between performance-oriented evaluation and query-dependent optimization.

Through comprehensive experiments across 8 benchmarks, we demonstrate that the proposed evaluator achieves an 83.7% validation accuracy in predicting prompt performance, surpassing embedding-based baselines by a large margin. When integrated into the optimization process, it consistently outperforms both static-template methods and query-dependent baselines across diverse reasoning and knowledge-intensive tasks. The evaluator further exhibits strong generalization to unseen domains such as MedQA and maintains stable optimization performance across different base models, underscoring its model-agnostic and portable nature.

Overall, our results validate that effective prompt optimization can be achieved without retraining the underlying LLM or relying on costly multi-execution feedback. By decomposing prompt quality into interpretable metrics and coupling evaluation with optimization, our framework provides a scalable foundation for future execution-free, performance-oriented, and cross-model prompt optimization in both standalone and multi-agent systems.

## 6 Limitation and Discussion

While the proposed framework provides a systematic and interpretable foundation for performance-oriented prompt optimization, several limitations remain that point to directions for future work.

(1) For the MATH500 benchmark, we report results only for GPT-4o. This dataset requires multi-step symbolic computation that exceeds the numerical and algebraic capabilities of small-scale models such as LLaMA-3-8B and LLaMA-3.1-8B. In these settings, most errors arise from the backbone model's inability to carry out the required calculations, rather than from prompt design itself, making prompt optimization largely ineffective. More generally, our results indicate that prompt optimization cannot fundamentally overcome the inherent capability limits of an LLM. Across all datasets, the performance gap between different backbone models, such as LLaMA-8B versus GPT-4o, is substantially larger than the improvement provided by any prompt-optimization method. Our evaluator therefore offers consistent gains within a model's natural capacity range, but its impact remains bounded by the underlying model's expressive and reasoning limits.

(2) The current evaluation framework focuses primarily on performance-related metrics. Although these metrics effectively capture the correlation between prompt quality and task accuracy, they do not account for other important dimensions such as safety, token efficiency, or user-centric factors (e.g., readability or controllability). Incorporating these broader criteria would enable a more comprehensive assessment of prompt quality in practical applications.

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

## A   Appendix

| Static Prompt Templates | Source |
|---|---|
| "Let's think step by step:" | zero-shot CoT (Kojima et al., 2022) |
| "Let's work this out in a step by step way to be sure we have the right answer:" | APE Zhou et al. (2023) |
| "First, decompose the question into several sub-questions that need to be solved, and then solve each question step by step:" | Least-to-most (Zhou et al., 2022a) |
| "Imagine three different experts are answering this question. All experts will write down 1 step of their thinking, and then share it with the group. Then all experts will go on to the next step, etc. If any expert realizes they're wrong at any point then they leave." | Tree-of-thought (Hulbert, 2023) |
| "3 experts are discussing the question, trying to solve it step by step, and make sure the result is correct:" | multi-agent debate (Liang et al., 2023) |

Table 5: Widely used static prompt templates for training dataset construction.

| Prompt Style | LLM Generated Prompt |
|---|---|
| Step-by-step | Solve the logic puzzle by explicitly defining truth values for each person, translating each statement into formal constraints, examining all logically possible assignments, eliminating inconsistent ones, and concluding with a clear determination of whether Vina tells the truth. |
| Expert Discussion | Simulate a panel of expert logicians who each contribute structured reasoning, challenge potential inconsistencies, refine the shared logical model of the puzzle, and collaboratively reach a consensus on whether Vina is telling the truth. |
| Socratic | Reason through the puzzle by asking yourself iterative Socratic questions that clarify assumptions, test alternative truth-value assignments, probe contradictions, refine provisional conclusions, and justify whether Vina tells the truth. |
| Creative | Analyze the puzzle using analogy or imaginative framing to highlight the dependencies among speaker claims, explore consistent scenarios, reconcile intuitive and formal reasoning, and end with a precise judgment about whether Vina tells the truth. |
| Verification | Generate a provisional answer to the puzzle, evaluate the internal consistency of all statements under that assumption, consider alternative assignments, revise if necessary, and conclude definitively whether Vina tells the truth. |
| Contrastive | Construct and compare multiple hypotheses about Vina's truthfulness, analyze their logical consequences for all speakers, reject inconsistent possibilities, and select the hypothesis that remains coherent. |

Table 6: Comparison of LLM-prompt-generation styles using a sample question from the BBH Web of Lies: "Rashida lies; Fletcher says Rashida lies; Antwan says Fletcher lies; Willian says Antwan tells the truth; Vina says Willian tells the truth; question: Does Vina tell the truth?" illustrates how different LLM prompting paradigms lead to distinct prompt formulations.

| | | |
|---|---|---|
| **Parent 1**
*(Expert Discussion)* | Segment 1: Simulate a panel of expert logicians who contribute structured reasoning | Hybrid Prompt: Simulate a panel of expert logicians who contribute structured reasoning, then probe contradictions, refine conclusions, and justify whether Vina tells the truth. |
| | Segment 2: challenge inconsistencies, refine the shared logical model, and reach a consensus on whether Vina tells the truth | |
| **Parent 2**
*(Socratic)* | Segment 1: Ask yourself iterative Socratic questions that test assumptions | |
| | Segment 2: probe contradictions, refine conclusions, and justify whether Vina tells the truth | |

Table 7: Illustration of the evolutionary recombination process using two prompts from distinct prompting styles, using the same example as Table 6. Each parent prompt is decomposed into two semantic segments, which are cross-combined to produce a hybrid prompt that blends features from both parents.

