# OpenReview forum: "Knowing How to Edit: Reliable Evaluation Signals for Diagnosing and Optimizing Prompts at Query Level"
_TMLR — Withdrawn by Authors_

### Review · Reviewer_AQg5 · 2026-02-26

**Summary Of Contributions:**

This paper studies prompt optimization as a process guided by performance-relevant evaluation signals. It designes a method to train a metric-based evaluator that predicts prompt quality directly from text without execution which bridges the gap between performance-oriented evaluation and query-dependent optimization. The emperical results show that the proposed evaluator achieves 83.7% accuarcy in predicting prompt performance. When incorporated into the optimization process, the results show that the proposed method outperforms existing optimization baselines across eight benchmark datasets on three backbone models.

**Audience:**

Yes

**Audience Explanation:**

1. Prompting has become a crucial way for people to leverage strong AI models. The execution-free evalutor proposed by this work provides a qualitative way of analyzing the quality of a prompt that goes beyond single query outcome, which is a meaningful contribution to the field.
2. The curated training data and a collection of eight datasets for evaluation provide a useful resource for the study of prompt optimization.

**Broader Impact Concerns:**

No, I don't see any ethical concern of this work.

**Claims And Evidence:**

No

**Claims Explanation:**

1. Although the paper claims that "the results show that the proposed method outperforms existing optimization baselines across eight benchmark datasets", I am unsure if the improvement is significant since the paper does not report statistical significance or confidence interval. For example, when we compare the proposed method and TextGrad, I think most of the results are on par rather than having improvements with a great margin. However, the proposed method adds more complexity in practice compared to TextGrad.
2. The method and findings in this paper are largely based on the feature important of prompt evaluation metrics from XGBoost (Table 1). This set of importance is computed with the prompts curated in Sec 3.1. It's unclear whether the same importance would transfer across different domains, especially given that the performance gains of the proposed method differ a lot among 8 datasets tested.

**Requested Changes:**

1. Report the statistical significance of the results in Table 4.
2. Conduct analysis on whether the set of importance fit with XGBoost on the curated training data is robust in different scenarios.

---

### Review · Reviewer_VcTa · 2026-04-14

**Summary Of Contributions:**

The paper proposes an evaluation-instructed prompt optimization framework that bridges the often-separate concerns of prompt evaluation and prompt optimization. The key contributions are:
- A performance-reflective evaluation framework that integrates four complementary prompt quality metrics (NLL, stability, MI, query entropy) selected via XGBoost feature importance.
- An execution-free evaluator built on a LoRA-fine-tuned LLaMA-8B that jointly performs binary classification (good/bad prompt) and metric regression, achieving 83.7% validation accuracy.
- A metric-aware optimization loop that uses gradient attribution from the evaluator to direct targeted, interpretable prompt rewrites at the query level.

**Key Strengths**:
- Clear motivation and principled connection between evaluation and optimization.
- Broad empirical coverage. The paper uses 8 datasets and 3 backbone LLMs, and compares the proposed method with multiple baselines.
- The optimization mechanism is interpretable and grounded in measurable metrics.

**Key Weaknesses**:
- Evaluation accuracy (83.7%) is reported on a validation split of the training distribution, rather than a held-out benchmark. This may raise concerns on whether the trained prompt quality predictor can generalize to other distributions.
- Several claims (e.g., "the first prompt evaluation–optimization pipeline") lack sufficient novelty justification, and hence risks over-claiming.
- Missing statistical significance tests and confidence intervals in all reported results. This calls into question as to whether the results are robust.
- Many results reported in Table 4 in fact do not show significant improvement over baselines. In addition, it would easier for visualization if another row is added at the bottom to show the averarage score of each method across all datasets.
- Another more important concern with the experimental results is that the compared baselines are in fact not very strong. For example, it would make the empirical results more convincing if more recent and stronger baselines are added, such as PromptBreeder ("Promptbreeder: Self-Referential Self-Improvement Via Prompt Evolution"), HbBoPs ("Hyperband-based Bayesian Optimization for Black-box Prompt Selection"), and MIPRO ("Optimizing Instructions and Demonstrations for Multi-Stage Language Model Programs").

**Audience:**

Yes

**Audience Explanation:**

Prompt optimization is a highly active and practically relevant research area. The idea of using structured, interpretable evaluation metrics to guide optimization, rather than relying on opaque LLM feedback, is a valuable and interesting direction. The cross-model generalization results and the execution-free evaluator design address real practical bottlenecks. Even with the evidentiary concerns noted above, the paper raises important questions and proposes mechanisms that would be of interest to researchers working on prompt optimization and potentially beyond.

**Broader Impact Concerns:**

Given that the work automates prompt optimization and could be used to systematically manipulate LLM outputs at scale (e.g., in adversarial attack contexts), a brief statement addressing potential misuse (e.g., optimizing prompts to bypass safety guardrails or produce misleading outputs) should be added.

**Claims And Evidence:**

No

**Claims Explanation:**

1. The headline claim of 83.7% accuracy is measured on an 80/20 split of the *same* training distribution. No independent test-set evaluation of the evaluator itself is reported, making it unclear whether this figure reflects genuine generalization or in-distribution fitting.
2. Several results in Table 4 show gains of only 0.01–0.02 over baselines (e.g., bbh_disambiguation_qa on llama3: 0.65 vs. 0.65 for Self-Refine; GPQA Diamond across all models: near-chance-level performance with differences of 0.01–0.02). No statistical significance tests or confidence intervals are provided, so it is impossible to determine whether reported gains are reliable.
3. The paper uses a maximum of 3 optimization iterations for all methods, but does not discuss whether this equally constrains all baselines or if the proposed method benefits disproportionately from the metric-guided step. Please give more explanation on the fairness of comparisons.
4. The paper claims to propose "the first prompt evaluation–optimization pipeline", but closely related work such as QPO and Prompt-OIRL also combine evaluation signals with query-dependent optimization. The distinction is not sufficiently argued.

**Requested Changes:**

1. Report the prompt quality prediction accuracy on a truly held-out test set, separate from the training distribution. The current validation split (yielding the 83.7% accuracy) may not be sufficient to support the generalization claims made.
2. Provide confidence intervals or significance tests for all performance comparisons in Table 4. Many reported gains are 0.01–0.03 and cannot be interpreted without confidence intervals.
3. More rigorously differentiate the proposed pipeline from QPO and Prompt-OIRL, which also use evaluation-guided query-level optimization. The "first" claim must either be substantiated or removed.
4. Report sensitivity of results to the number of optimization iterations across methods to ensure fairness.
5. The red bars in Figure 2 (e.g., bbh_web_of_lies on LLaMA-3.1) indicate performance drops and a failure case. This should be analyzed and explained.

---

### Review · Reviewer_v92i · 2026-04-16

**Summary Of Contributions:**

The paper proposes an evaluation-instructed prompt optimization framework. First, the paper trains an execution-free evaluator to predict prompt effectiveness (accuracy and 4 other relevant metrics) directly from query and prompt, eliminating the need to run model executions for prompt optimization. These evaluation signals are then used to guide targeted and interpretable prompt updates. Empirically, the approach achieves strong predictive accuracy and consistently outperforms existing prompt optimization baselines.

**Audience:**

Yes

**Audience Explanation:**

Yes, a prompt optimization method which does not require rollouts from the model seems generally useful to a broad audience. My main concerns are not with the applicability of this work, rather some of the design decisions and clarity of technical detail.

**Claims And Evidence:**

No

**Claims Explanation:**

The claims are generally supported and the paper is well-written. The method to train the execution-free evaluator is described in detail, the design decisions are sound and the evaluator achieves high validation accuracy. However, the actual evaluator-guided prompt optimization process is not defined formally and is vague. It is unclear how signal from the evaluator is used to actually make text updates to the prompts. It is also unclear how features of the prompt affect these performance metrics (concern detailed in requested changes section). I also feel that there are some major concerns in evaluation if I am reading the results right (refer to the requested changes section for my questions).

**Requested Changes:**

- How is the prompt optimization formally done in practice? Can the authors provide pseudocode or some formal explanation of the process. Can the authors show some examples of what the initial and final prompt looks like for some queries?
- The evaluator can be interpreted as a reward function. Prompt optimization tries to find a prompt that is scored highly by the evaluator. However, doing multiple iterations of this might lead to reward hacking (evaluator scores a prompt highly but it actually performs poorly). Do authors find any instance of this?
- I am a bit concerned with Table 4 metrics. If I am reading the table correctly, it seems that the performance of GPT-4o is on par with that of Llama-8B on multiple tasks. This should not be the case and suggests that there is some issue in the pipeline. I am also concerned by the fact that Math500 results for Llama are not reported. I understand that smaller models are bad at Math, but numerous papers (including the Llama model release) report performance on this dataset. It is surprising that the authors chose to exclude this.
- When describing performance metrics for the evaluator, the authors present properties of a prompt that result in a low score for the metric (example, poor NLL is caused instruction conflict, noisy or long preambles, etc). How do the authors obtain these features? There is no experiment shown to determine this - it seems hand wavy.

---

### Note · Authors · 2026-05-19

I have read and agree with the venue's withdrawal policy on behalf of myself and my co-authors.